# Rapid Actions of the Nuclear Progesterone Receptor through cSrc in Cancer

**DOI:** 10.3390/cells11121964

**Published:** 2022-06-18

**Authors:** Claudia Bello-Alvarez, Carmen J. Zamora-Sánchez, Ignacio Camacho-Arroyo

**Affiliations:** Unidad de Investigación en Reproducción Humana, Instituto Nacional de Perinatología-Facultad de Química, Universidad Nacional Autónoma de México, Ciudad de México C.P. 0451, Mexico; clautus.bello@gmail.com (C.B.-A.); carmenjaninzamora@comunidad.unam.mx (C.J.Z.-S.)

**Keywords:** nuclear progesterone receptor, cSrc, rapid actions, breast cancer, glioblastoma

## Abstract

The nuclear progesterone receptor (PR) is mainly known for its role as a ligand-regulated transcription factor. However, in the last ten years, this receptor’s extranuclear or rapid actions have gained importance in the context of physiological and pathophysiological conditions such as cancer. The PR’s polyproline (PXPP) motif allows protein–protein interaction through SH3 domains of several cytoplasmatic proteins, including the Src family kinases (SFKs). Among members of this family, cSrc is the most well-characterized protein in the scenario of rapid actions of the PR in cancer. Studies in breast cancer have provided the most detailed information on the signaling and effects triggered by the cSrc–PR interaction. Nevertheless, the study of this phenomenon and its consequences has been underestimated in other types of malignancies, especially those not associated with the reproductive system, such as glioblastomas (GBs). This review will provide a detailed analysis of the impact of the PR–cSrc interplay in the progression of some non-reproductive cancers, particularly, in GBs.

## 1. Introduction

Progesterone (P4) is one of the most studied and characterized female sex hormones in the scenario of cancer [1,2,3]. P4 actions can be exerted by a diverse group of receptors, including nuclear progesterone receptors (PRs), membrane progesterone receptors (mPRs), and membrane-associated progesterone receptor components (PGRMCs). The mechanisms of P4 actions are classified as genomic or non-rapid effects when involving the transcription of P4-responsive genes and non-genomic or rapid effects when P4 effects are mediated by signaling through cytoplasmatic proteins [4,5].

Although the PR is primarily known for its function as a ligand-activated transcription factor, its interaction with P4 also triggers rapid or transcription-independent effects. Immediate effects mainly occur through the activation of mPRs [6] and PGRMCs [7,8,9], whereas nuclear actions are exerted by the PR [10], but the latter is the only one that can exert both effects. In addition to the domains involved in its function as a transcription factor, the PR possesses a polyproline-rich (PXPP) motif between aa 421 and 428 that binds to the SH3 domains of several cytoplasmic molecules, including cSrc (Figure 1), hematopoietic cell kinase (HCK), Fyn, and other kinases or adapter proteins such as the regulatory subunit of PI3K (p85), CRK proto-oncogene adaptor protein (Crk), and growth factor receptor-bound protein 2 (Grb2) [11]. cSrc is one of the most well-studied and well-characterized non-receptor tyrosine kinases in cancer progression. Unlike other proteins associated with the hallmarks of cancer, cSrc has no mutations but exhibits high enzymatic activity [12].

PR–cSrc interaction has mainly been studied in breast cancer. Since the late 1990s, a large body of evidence has accumulated about the effect of P4 on breast cancer cells through rapid PR actions [13,14]. In several breast cancer-derived cell lines, it has been described that once stimulated by P4, the PXPP motif of the PR binds to the SH3 domain of cSrc, promoting a conformational change in this kinase that exposes its autocatalytic domain, followed by its activation (Figure 1) [15,16]. The interplay between the PR and cSrc in breast cancer leads to the activation of signaling pathways involved in proliferation (ERK-MAPKs) (19), migration, and invasion (focal adhesion kinase (Fak)—focal adhesion complexes) (Figure 1) [17,18]. In breast cancer, some evidence suggests that the nuclear estrogen receptor (ER) mediates the PR–cSrc interaction [13,14]. However, many elements in the mechanism of cSrc activation through the PR remain to be elucidated, for example, whether other proteins stabilize the PR–cSrc interaction. Another aspect of interest is to elucidate the PR regulation by cSrc. In this review, we aimed to discuss these aspects, the unknown processes in this mechanism, and the impact on cancer progression.

## 2. SH3 Domain–PXPP Motif Interaction: Structural Basis and Functions

The protein–protein interactions mediated by SH3 domain–PXPP motifs are one of the most abundant and studied processes in cells since they are necessary for activating signaling pathways and for protein subcellular localization. Although the SH3 domain was first described as an extra-catalytic domain of the Src family kinases (SFKs) in the 1980s, they are present in a wide variety of proteins: other tyrosine kinases such as the Abl family and cytoskeletal proteins such as actin and myosin [19,20]. Around 300 SH3 domains have been identified in the human genome and in more than 200 different proteins [21].

In the case of SFKs, the protein–protein interaction mediated by the SH3 domain promotes their activation. Once activated, such kinases regulate diverse signaling pathways whose final effect is inducing cellular proliferation, cell survival, and migration, among other effects. Their broad cross-talk with many different transduction pathways makes them key regulators in pathologies such as cancer [22].

The SH3 domains consist of approximately 55–85 amino acids with a conserved structure folding [23]: five to six β-strands arranged as antiparallel β-sheets or as β-barrels connected by three loops and one helix. In addition, the SH3 domain is rich in aromatic amino acid residues that stabilize the binding site interaction with their ligand. These last elements are relevant to peptide ligand recognition [24]. The canonical ligands of SH3 domains are the PXPP left-handed helices [25].

The PXPP motifs are highly abundant in the human proteome [26]. Such protein motifs have a pseudo-symmetrical structure, which could be recognized in two different orientations by the PXXP-binding site of the SH3 domain. There are two classes of the PXPP motifs ligands which differ in the consensus sequence orientation: The class I consensus sequence is RXLPPXP, whereas class II is constituted of the consensus sequence XPPLXPR (the opposite orientation from class I) [27]. Positive amino acids such as arginine and lysine are necessary for the recognition of the polyproline motif by the SH3 domain. Such residue is recognized by the specificity pocket of the SH3 domain, formed by negative amino acid residues adjacent to the PXPP motif binding site [28,29]. In the particular case of SFKs, it was reported that they bind to the class I consensus sequence of polyproline motifs [27].

The Src kinases family is constituted by nine members whose structural organization is highly conserved in humans. Near their N-terminal domain (NTD), the SH4 domain is located, and its post-translational modifications such as myristoylation and palmitoylation are involved in attaching the kinase to the cell membrane. The SH4 domain is also one of the most variable regions among SFKs. Aside from SH4, the SH3 and SH2 domains regulate protein–protein interactions and the catalytic activity of the Src family. Interactions mediated by SH3 domain–PXPP motifs are key regulators in the function of SFKs. The SH2 is attached with a linker section to the SH1 domain, which is the enzyme’s catalytic center. It is followed by the C-terminal short section containing an autoinhibitory phosphorylation residue.

When the C-terminal residue is phosphorylated, it remains bound to SH2, maintaining Src in an inactive conformation. The SFKs will change to their active form if dephosphorylation of such residue occurs or if a protein–protein interaction occurs in the SH2 or SH3 domains and induces their conformational change [30]. Many protein substrates can recruit and activate SFKs by interacting with the SH3 and/or SH2 domains. Some ligands of Src are growth factor receptors, integrins, other kinases such as Fak, and intracellular steroid receptors such as the PR (Figure 1) [31,32].

The PR belongs to the superfamily of nuclear receptors, which are best characterized by their function as transcription factors. The PR is part of the nuclear receptor subfamily 3, which comprises other steroid receptors such as ER, androgen (AR), glucocorticoid, and mineralocorticoid receptors. This family of receptors has very high variability in the NTD, favoring their interaction with specific proteins such as coactivators, other nuclear receptors, and hub proteins of different signaling pathways such as the Src family [33]. Two main PR isoforms in humans have been reported: PR-A and PR-B. Although their transcription is regulated by two distinct promoters, they are coded by the same gene (11q22–23). Structurally, the PR-B is the longer isoform and has 164 more amino acids than the PR-A in the NTD region [34].

The membrane or cytoplasmatic localization of steroid receptors is crucial to their participation in activating rapid non-transcriptional pathways. In the case of the PR, ER, and AR, their attachment to caveolae lipid rafts of the plasma membrane is mediated by their palmitoylation at the ligand-binding domain (LBD) [35,36]. This post-translational modification is promoted by the heat shock protein 27 (Hsp27) in the ER [36]. Once attached to the plasmatic membrane, steroid receptors can interact with other proteins of the focal adhesion complexes. These stable structures at the plasma membrane interact with extracellular matrix components that mediate cellular responses to the external and inner signals regulating metabolic activity, proliferation, and motility [35].

In the PR-B, the PXPP motif is located between the 421 and 428 amino acids in the NTD region. When a direct interaction exists between cSrc and the PR, the former transits to its active form and promotes the activation of other kinases such as the mitogen-activated protein kinases (MAPKs) [16] (Figure 1). The activation of such signaling cascades has high repercussions in the cells, particularly, in different cancers, as described in the incoming sections (Figure 1). It is also important to mention that other steroid receptors also promote the activation of such signaling cascades, although their interaction with cSrc proteins could be different from the PR. The interaction between the AR and cSrc is also mediated by the interaction of SH3–PXPP, but the ER interacts with this kinase at the SH2 domain [37,38].

## 3. Functions of the Polyproline Motif of PR in Breast Cancer

Most research about rapid PR actions has been conducted on hormone-dependent cancers. Most PR rapid effects depend on the activation of cSrc by SH3–PXPP motif interactions or through its indirect interaction mediated by the ER [13,14,15].

In the first half of the 1990s, the molecular mechanisms involved in breast cancer progression by the action of sex hormones were still unknown. In 1998, Migliaccio et al. first reported progestin-dependent activation of cSrc and that this effect was dependent on the formation of the ER–cSrc–PR complex [13]. Interestingly, three years later, Boonyaratanakornkit et al. reported that activation of cSrc by the synthetic progestin R5020 was not dependent on the presence of the ER and that the PR could directly interact through the PXPP motif–SH3 domain [15]. Subsequent studies have shown that the PR and ER can directly interact with cSrc; however, the mechanisms that allow this interaction are specific for each receptor [11]. The dynamic of the interaction in cells expressing both receptors is not known. Answering in which contexts the participation of the ER is essential for activation of cSrc through the PR (Figure 2) would provide valuable information for the development of more effective therapies against breast cancer.

In vitro models of breast cancer have enlightened the relevance of PR interaction with kinases attached to the plasma membrane. The cell proliferation enhanced by progestins has been extensively reported in breast cancer [39]. In the T47D breast cancer cell line, Skildum et al. demonstrated that the mutant form of the PR-B (S294A PR-B) with low transcriptional activity and diminished proteasomal degradation [40] could activate the Ras/Raf/MAPK cascade in a ligand-dependent manner. In addition, the S294A PR-B and wt-PR induce cell cycle progression [41]. This is interesting considering that only about 5% of the PR is at the plasma membrane [15]. The direct cSrc activation by the PR is as fast as 5 to 10 min, and apart from MAPK activation, such effects activate ERK and EGFR, which in T47D also activate transcription factors such as Sp1, which promotes the expression of genes without PRE, like p21 protein [41,42].

Moreover, the positive feedback between PR rapid signaling and growth factor-activated pathways was reported in breast cancer cell lines. The co-localization of EGFR, PR, and cSrc has been reported in the focal adhesion complexes. In this context, the activation of EGFR depends on the ligand-dependent activation of the PR-B [42]. In addition, the Jak/STAT pathway is activated by PR–cSrc in T47D and the mammary-induced tumor model with C4HD cells. In such models, the progestin medroxyprogesterone induces the rapid activation of cSrc with the subsequent activation of the signal cascade Jak1/Jak2/STAT3.

Additionally, in the longer term (48 h), medroxyprogesterone also promotes an augment in STAT3 levels [43]. In T47D breast cancer cells, P4 enhances breast cancer cell migration and invasion via activating Fak through extranuclear actions of the PR [17]. The Fak kinase also contains an SH3 domain that mediates the interaction with several of the components of focal adhesions, including cSrc [44]. However, it is unknown whether the SH3 domain of Fak has an affinity for the PXPP motif of the PR and its role in PR localization in the plasma membrane near the focal adhesion complexes.

## 4. Role of PR and cSrc in Glioblastoma Progression

### 4.1. Contribution of cSrc to Malignancy of Glioblastoma

Glioblastoma (GB) is the most frequent and aggressive malignant brain tumor in adults. The current standard of care for patients with GB does not offer a survival of more than 15 months [45]. cSrc is one of the oldest proto-oncogenes associated with the progression of cancer. For a long time, it was thought that the central role of this kinase (cSrc) was related to cellular proliferation and tumor growth. However, in the last two decades, interest in this kinase has emerged in the context of cell adhesion, invasion, and motility [12]. cSrc activity has been reported to be higher in GBs than in normal brain cells [46,47]. One of the first studies on the role of cSrc in GBs was performed in transgenic mice constitutively expressing the mutated variant of cSrc lacking the negative regulatory domain at the C-terminal end (v-SRC). In these animals, glial tumors grew with molecular and morphological characteristics closely resembling those of a human GB [48]. The role of cSrc in regulating GB cell motility was reported by Angers-Loustau et al. in 2004. They implanted spheroids of the human GB cell line (U251) in three-dimensional type I collagen matrices. It was observed that specific pharmacological inhibitors of the Src family, PP2 and SU6656, significantly reduced cell invasion. In addition, PP2 interfered with actin filament rearrangement in lamellipodia formation [49]. In 2015, Lewis-Tuffin et al. found that the silencing of cSrc, Fyn, Yes, and Lyn decreased proliferation and migration in human GB-derived cell lines LN229, U87, U251, TP483, and SF767 [50]. In GB cells with positive expression of the stemness marker CD133, the inhibition of expression and activity of cSrc and Fyn decreased these cells’ migratory and invasive capacity [51]. In 2017, it was shown that in patient-derived GB (SOX2+/Nestin+) stem cells, the addition of a penetrating peptide, whose sequence corresponded to amino acids 266–283 of the connexin 43 sequence, significantly reduced the motility and invasive capacity of these cells through the inhibition of cSrc and Fak [52]. Although the evidence suggests that cSrc is a potent drug target, clinical trials with cSrc inhibitors have not been satisfactory [53]. One of the factors analyzed is the failure of Src inhibitors to cross the blood–brain barrier. In this regard, it may be helpful to address proteins that participate in the activation of cSrc which have inhibitors or antagonists that can cross the blood–brain barrier.

### 4.2. PR: An Underappreciated Villain in GB Progression

Since 1997, several studies have correlated PR content with gliomas’ malignancy. Khalid et al. found that the protein content of the PR was higher in GBs (grade IV gliomas) than in grade I and II gliomas from the biopsies of 86 patients [54]. Recently, Arcos-Montoya et al. reported that the PR content, determined with immunofluorescence, was higher in samples derived from patients with GBs than in samples from patients with lower-grade gliomas or normal tissue [55]. In addition to the positive correlation between protein expression and grade of malignancy in gliomas, there is functional evidence of the PR’s role in the progression of GBs. Gonzalez-Aguero et al. treated human GB-derived cell lines with P4 (10 nM) and observed a significant increase in the proliferation rate compared to the vehicle. When RU486, a PR antagonist, was added to the cells, the effect of P4 was blocked, suggesting that this hormone induces GB cell proliferation through the PR [56]. The effects mediated through PRs on GB cells are not limited to modifying proliferation-associated events. Piña-Medina et al. found that P4 increased the migratory and invasive capacity of the GB-derived cell line, U251. This effect was partially blocked when RU486 or antisense oligonucleotides against the expression of PRs were added [57]. However, these effects have generally been associated with the transcriptional activity of the receptor without considering its extranuclear role in the activation of signaling pathways in the cytoplasm. In breast cancer, PR extranuclear functions have been extensively characterized; however, in other types of tumors not associated with the reproductive system, such as GBs, this aspect has been underestimated.

### 4.3. PR–cSrc Interaction in GBs: Experimental Evidence and Future Perspectives

In breast cancer, the interplay between PR and cSrc has been widely characterized [13,14,15]; however, in other non-reproductive cancer, knowledge about this phenome is scarce. In the GB context, accumulating evidence suggests that the PR participates in the progression of the tumors when stimulated with low concentrations of P4 [56,57,58,59]. However, these effects have been primarily associated with transcriptional activity [60]. Recently, Bello-Alvarez et al. identified rapid effects induced by low P4 doses (50 nM) in GB-derived cell lines. Firstly, P4 induced cSrc activation after ten minutes of stimulation. When a siRNA against PR expression was added, this effect was abolished, suggesting the role of the PR in the kinase activation. In addition, P4 activated Fak after twenty minutes of treatment, and this effect was partially suppressed when a siRNA against cSrc expression was incorporated. These results suggest that once stimulated by P4, the PR induces rapid signaling in GB-derived cell lines through the activation of cSrc and Fak, both essential proteins that regulate the focal adhesion complex assembly and disassembly, and contribute to migration and invasion.

The authors also reported the PR–cSrc interaction by co-immunoprecipitation assay [59]. In summary, the findings of this publication propose that in GBs, the PR operates via a rapid mechanism as in the case of breast cancer [61]. Nevertheless, there are still many missing elements in this mechanism that need to be elucidated. For example, the activation of other Src family proteins by the PR (Figure 2), especially Lyn, whose activity is the highest in GBs [62]. According to Boonyaratanakornkit et al., in breast cancer cells expressing PRs and ERs, the activation of cSrc by P4 involves the formation of an ER–cSrc–PR ternary complex [11]. In GBs, the expression of both receptors and their participation in the progression of these tumors has been verified [63]. Hernández-Vega et al. demonstrated that E2 stimulated epithelial–mesenchymal transition, migration, and invasion of glioblastoma cells. Interestingly, these effects were only mediated by the ERα subtype [64]. It would be of great interest to determine the specific ER subtype that participates in the activation of cSrc through the PR in GB cells (Figure 2).

In breast cancer cells, it was reported that cSrc phosphorylates the Tyr537 residue of ERα, which in vitro and in vivo enhances ERα binding to EREs [65]. In silico analysis performed by Bello-Alvarez et al. proposed the residue Tyr87 of the PR as a putative site for cSrc phosphorylation [59]. This result encourages us to extend this field of research to find out which kinases of the Src family are involved in PR phosphorylation and their effects on the transcriptional activity of this receptor (Figure 2).

Kawprasertsri et al. reported in lung cancer cells without progestin stimulation that the rapid PR signaling interferes with the activation of the EGFR–ERK1/ERK2 pathway. Through its PXPP motif, the PR can bind to the adaptor protein GrB2, which is essential for signaling through EGFR. Although not demonstrated, these authors suggested that the PR limits the availability of GrB2 to EGFR [66], which has signaling that is one of the most studied phenomena in the context of GB and is known to contribute to its progression [67]. However, there is no information about the possible cross-talk between extranuclear PR activity and EGFR-mediated effects in GBs. This result is exciting because it highlights the possibility that the PR could promote glioblastoma progression depending on the type of SH3-domain interacting protein. These findings lead us to the following question: which other proteins with SH3 domains can interact with the PR, and what effects are triggered? (Figure 2).

## 5. Relevance of Possible cSrc–PR Interaction in Other Cancers

Lung and colorectum carcinomas are non-reproductive cancers with the highest incidence and mortality (Global Cancer Observatory—https://gco.iarc.fr/, accessed on 20 April 2022). As SRC is one of the most studied and characterized proto-oncogenes, the role of cSrc in the progression of both entities has been widely reported [68,69,70,71].

Non-small cell lung cancer (NSCLC) is the most prevalent form of lung cancer and is regulated by a complex signaling network [72]. In NSCLC, the role of cSrc is closely linked to EGFR. Zhang et al. reported that in two EGFR-dependent NSCLC cell lines (HCC827 and H3255), the phosphorylation of SFKs was higher than in non-EGFR-dependent cell lines. In both cell lines, treatment with the SFK inhibitors PP1 or SKI-606 induced apoptosis [73]. Although outside the context of non-reproductive cancers, the role of the PR is less well-known, there is evidence about its function in NSCLC. In contrast to cSrc, PR expression in NSCLC has been associated with a favorable prognosis [74,75]. Kawprasertsri et al. found the presence of the PR-B, but not that of the PR-BΔSH3 (PR variant with a mutation in the PXPP motif which inhibits the interaction with SH3 domains), and a decrease in EGF-induced A549 proliferation and ERK1/2 activation, suggesting the role of the extranuclear function of the PR through its PXPP motif in EGFR signaling [66]. cSrc is one of the main downstream regulators of EGFR signaling [71]. The PR has been reported to participate in EGFR transcription [60,76]. It would be interesting to evaluate whether there is a cooperation among cSrc, EGFR, and PR. A possible scenario would include the activation of cSrc by its interaction with EGFR, and once activated, the phosphorylation of different tyrosine residues in the PR, which could modify its transcriptional activity.

cSrc has been extensively investigated in colon cancer. Evidence suggests that in this malignancy, instead of inducing proliferation, cSrc promotes the assembly of integrin adhesions, strengthening the ability of cells to spread on a substrate. The proposed mechanism regulates focal adhesions through its interaction with integrins and other focal proteins, including paxillin [70]. In colorectal cancer cell lines, it was found that the PR induced folic acid-mediated antiproliferative effects through cSrc activation [77]. Considering that P4 effects differ according to the concentration used, favoring (10–50 nM) [57,59] or decreasing (80–300 µM) [78,79] GB progression, it would be of great interest to evaluate its effects at different concentrations (Figure 2) on PR extranuclear functions in non-reproductive cancers.

Sex steroid receptors, including the PR, have been located in the plasma membrane. Among the events regulating this phenomenon is palmitoylation in the ligand-binding domain. In the case of the ER, the heat shock protein Hsp27 is essential for membrane tethering [36], but in the case of the PR, the proteins involved in translocation and anchoring to the membrane have been poorly studied. Another new aspect, even in breast cancer, is whether the PR needs to be associated with the plasma membrane to initiate signal transduction and the complex of proteins associated with the PR once it is anchored to the membrane.

## 6. Conclusions and Perspectives

In addition to its role as a transcription factor, the PR activates signaling cascades in the cytoplasm through its PXPP motif with the SH3 domain of a diverse group of proteins. This mechanism has been widely studied in breast cancer cells [11,13,14,15]. However, its potential involvement in the progression or arrest of other malignancies has been underestimated. In the context of the rapid actions of the PR, there remain numerous unanswered questions. First, it is unknown which proteins with SH3 domains interact with the PR and the effect of this interaction. It is unknown how the PR’s stability and transcriptional activity can be regulated through its phosphorylation at different tyrosine residues by the SFKs. Another aspect to consider is to evaluate the effect of P4 concentration in PR extranuclear actions. In in vitro and in vivo models of prostate cancer, the addition of a PXPP peptide targeting the AR–cSrc association decreased cell proliferation in the LNCaP cell line and tumor growth in mice compared to the effects of the control peptide [80]. This result suggests the potential of this type of strategy for cancer treatment and the importance of extending the study of the rapid mechanism of the PR and other members of the nuclear receptor family to non-reproductive cancers.

## Figures and Tables

**Figure 1 cells-11-01964-f001:**
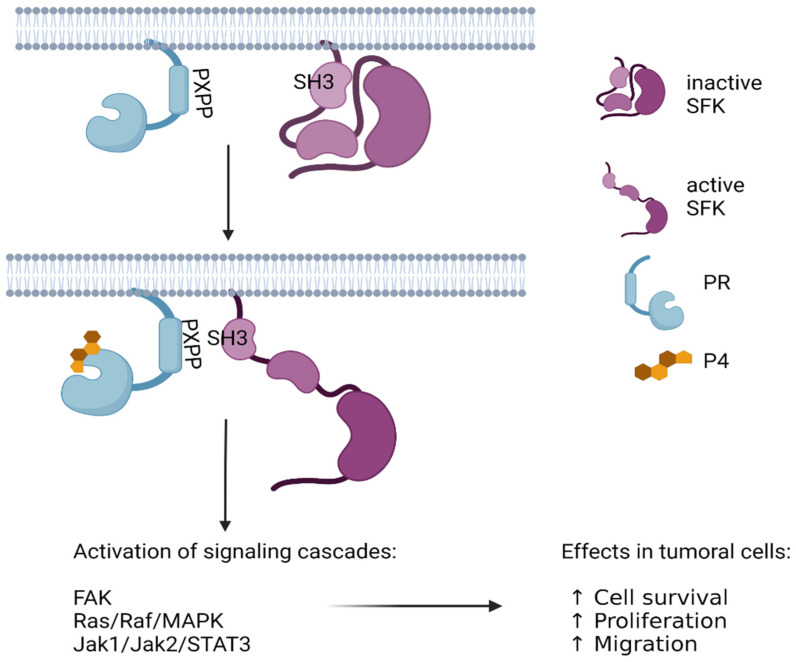
Activation of SFKs by SH3–PXPP interaction. In the cytoplasm or even anchored to the plasma membrane, PR interacts with other cytoplasmic molecules leading to the activation of various signaling cascades. Via PXPP, this receptor interacts with the SH3 domains of several molecules such as cSrc kinase. A direct interaction between cSrc and PR causes a conformational change in cSrc towards its active form and promotes the activation of other signaling cascades such as mitogen-activated protein kinases (MAPKs) that favor cancer progression by enhancing cell survival, proliferation, and migration.

**Figure 2 cells-11-01964-f002:**
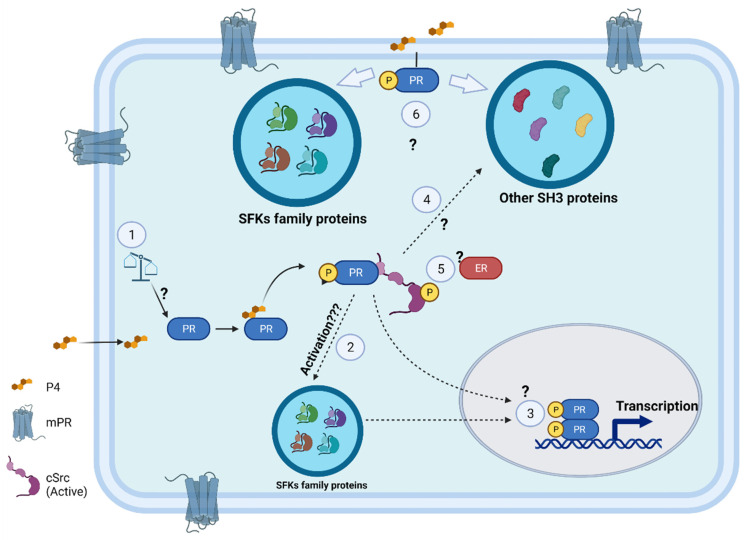
Potential extranuclear effects of PR. In breast cancer cells, cSrc kinase has been reported to bind PR through SH3–PXPP interaction. This interaction promotes the activation of cSrc by a conformational change that exposes the tyrosine residue 416. Other SH3-domain proteins could interact with the PR. Although its main localization is nuclear and cytoplasmic, the PR has been found to be anchored to the plasma membrane. However, there is still a lack of knowledge about many of the proteins and effects involved in the extranuclear signaling of the receptor. 1: What role does P4 concentration play in the induction of rapid PR effects? 2: What other kinases of the SFKs family can be activated by their interaction with the PR? 3: What is the role of cSrc and other SFKs in PR phosphorylation? 4: Which other proteins with SH3 domains can interact with the PR, and what effects are triggered? 5: In what context is the ER essential for the activation of cSrc through the PR? 6: What are the mechanisms allowing PR anchoring to the plasma membrane? What protein complexes would form in this area? Furthermore, is this PR localization essential for extranuclear signaling?

## Data Availability

Not applicable.

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
