# Peer review of "Rapid Actions of the Nuclear Progesterone Receptor through cSrc in Cancer"

_cells, 2022, doi:10.3390/cells11121964_

Round 1

Reviewer 1 Report

The authors performed a comprehensive review on progesterone receptor action via cSrc in cancer. My main complaint is about the length of the article. The introduction is too detailed, too long, and provides much information which is not important taking into consideration the main aim of the article. Similarly, in other parts of the article, the authors mainly enumerate the list of findings from various studies without appropriate commentary on the consequences - such a paper should provide more authors' opinions. In figure 2 the authors make a list of potential research questions - I think that it should be moved to the body of the manuscript with adequate comments. Moreover, Figure 2 could have better esthetics. Finally, I think that the article should be proofread and improved by an English native speaker.

Author Response

Thank you very much for your valuable comments. We appreciate all the observations you made to improve the quality of our work. As you requested, we have worked on the length of this paper. In the "Introduction" and "SH3 domain-PXPP motif interaction: structural basis and functions" sections, we have removed information that did not directly contribute to the main content and objective of the paper. In the "Functions of the polyproline motif of PR in breast cancer" section, we added and discussed information relevant to the objective of the paper. In addition, we have incorporated more opinions on reported papers, the impact of their results, considering the questions that remain to be answered, and the importance this may have in the field of cancer therapies. As you suggested, Figure 2 was replaced by a more esthetic version. English was again revised and corrected. All modifications made to the manuscript are highlighted in yellow.

Reviewer 2 Report

I am glad to recommend this paper for publication. Namely, it is a relatively innovative analysis. I am of the opinion that such research is relevant to our field. I am of the opinion that this work will encourage further development of the field. The paper is well written, clear and focused. 

Author Response

Thank you very much for your kind opinion about our work. We are pleased to make this impression on you.

Round 2

Reviewer 1 Report

The authors improved the paper according to my comments. I especially appreciate the modification of "Figure 2". The paper is now suitable for publication.